# Static Torque Analysis of Micro Claw-Pole Stepper Motor Based on Field-Circuit Combination

**DOI:** 10.3390/mi13091517

**Published:** 2022-09-13

**Authors:** Yuanxu Xin, Yan Sun, Xudong Wang, Xiaofei Xi, Yabin Su, Yong Yang

**Affiliations:** School of Electrical Engineering, Shanghai Dianji University, Shanghai 201306, China

**Keywords:** micro, stepper motor, finite element analysis, the equivalent magnetic circuit method, electromagnetic torque

## Abstract

Because of the complexity of the structure and magnetic circuit of the micro claw-pole stepper motor, it is difficult to analyze this kind of motor quickly and accurately. Therefore, it takes a lot of time to accurately model and use the three-dimensional finite element analysis method to accurately analyze the motor. Regarding the three-dimensional finite element method, the equivalent magnetic circuit method analysis is fast, but the accuracy is not high. In order to better study the performance of this kind of micro claw-pole motor and reduce the cost of optimization time, this paper adopts the method of combining the equivalent magnetic circuit method and three-dimensional finite element analysis to analyze the static torque characteristics of the micro permanent magnet claw-pole stepper motor. Firstly, the equivalent magnetic circuit method is used for theoretical analysis, the air-gap flux equation is deduced, and the relationship between the electromagnetic torque and the geometric parameters of the motor is deduced. Then, the three-dimensional finite element simulation results are substituted into the relevant formulas defined by the equivalent magnetic circuit method to obtain a more accurate electromagnetic torque. Finally, through the comparison and analysis of the experimental data, simulation data, and theoretical calculation values, the error rate of the derived motor torque is within 8.5%. The micromotor studied in this paper is optimized, and the holding torque is increased by 12.5% under the premise that the braking torque does not change much. The simulation calculation time is effectively shortened, the analysis difficulty is reduced, and the calculation accuracy is high. It is shown that the method combining the equivalent magnetic circuit method and the three-dimensional finite element analysis method is suitable for preliminary design research and optimization calculation of the micro claw-pole stepper motor.

## 1. Introduction

Micro permanent magnet claw-pole stepper motors are widely used in virtual reality (VR) devices for the self-adjustment of interpupillary distance due to their small size, large torque, high positional accuracy, and low cost. Due to the complex magnetic circuit of this type of motor, magnetic flux leakage is common, in addition to complex, easy saturation of ferromagnetic materials, small torque, high difficulty in optimal design [1], and low assembly consistency. Therefore, many scholars are devoted to the miniaturization research of this motor and the performance optimization of the motor.

At present, three methods are commonly used for 3D electromagnetic field calculation, namely the analytical method, finite element method, and equivalent magnetic circuit method [2]. Different from ordinary motors, micro claw-pole stepper motors are very small in size, their structure and magnetic field distribution are typically three-dimensional, and the motor has a lot of magnetic flux leakage and is complex, which directly makes the design and analysis of this type of motor very difficult. Therefore, many scholars directly use the three-dimensional finite element method for calculation and analysis. Son, Y.K. used the 3D FEA method to analyze the holding torque and detent torque of the motor using the geometric parameters of permanent magnets and claw teeth to obtain more accurate modeling [3]. Ham, S.H. used the 3D FEA to analysis and believes that the assembly error is the main reason for the increase in the motor’s detent torque [4]. Jung, D. conducted a detailed finite element analysis of the design parameters of the motor stator poles by using the design of experiment (DOE) method, and studied in detail the tooth shape, the number of turns [5], and the permanent magnet overhang [6], as well as the rotor shape [7] and segmented permanent magnets to assess the influence on the static torque characteristics of the motor. Finally, the performance of the motor is improved by a trade-off study [8]. The three-dimensional finite element method can consider the influence of multiple effects at the same time, combined with powerful finite element calculation software, to achieve high-precision electromagnetic field analysis, but the analysis and calculation amount are large, the calculation time is long, and the computer hardware requirements are high [9]. Moreover, most research objects are larger size motors, or generators, and there is scarce research on such micro-motors. However, some parameters that are ignored in large-sized motors have become very important in miniaturization research. Therefore, the optimization design of such motors is very difficult, which also leads to the suboptimal configuration of micro-motors applied in different application scenarios. Applying motors with similar configurations to different products will result in motors of the same volume or different batches of motors having different performances and low yields. Therefore, it is necessary to find a method with high portability, high precision, and rapidity to simplify the process of motor optimization design.

In this research direction, the three-dimensional finite element analysis method is too time-consuming. Stuebig, C. adopted a method combining analytical-numerical calculation [10] to improve the calculation of different types of hybrid stepper motors for the problem. The method has acceptable accuracy, but there are still some discrepancies, which still need to be analyzed by magnetic circuit analysis. Gong, J.S. [11] deduced the torque formula by means of equivalent permeance and studied the influence of winding and permanent magnet material Br on the motor torque. However, error analysis of the derived formula was not carried out, and some magnetic flux leakage is ignored, resulting in a large error. Yeo, H. proposed an analytical method that combines the magnetic equivalent circuit model with analytical solutions of Laplace’s equation and Poisson’s equation [12], which greatly reduces the computation time. Through the analytical method, a simple magnetic field analysis and calculation can be performed under certain assumptions and simplified conditions. However, the analytical method is complicated to calculate and requires a lot of formula derivation. In addition, factors, such as magnetic circuit saturation, stator cogging, and magnetic flux leakage, are difficult to accurately reflect in the analytical formula, which also leads to a large final calculation error. In addition, some scholars have analyzed the magnetic circuit of the claw-pole stepper motor and derived the air-gap magnetic flux equation using the Thevenin equivalent circuit [13]. On the basis of magnetic circuit analysis, the characteristics of static torque and dynamic torque are given. However, the actual measured value of the maximum holding torque calculated according to the formula still has an error of 0.026 Nm. Rhyu, S. [14] studied the characteristics of the actual model by using the equivalent magnetic circuit method and the three-dimensional finite element analysis, respectively, and indicated that the magnetic flux leakage was the main reason for reducing the performance. However, there is a 60% error between the actual measurement and the 3-D finite element simulation results, and an 80% error with the equivalent magnetic circuit method solution. Considering the saturation of the entire rotor part based on the 3D geometry and material information, Kwon, S.O. established a 2D equivalent model of the claw pole motor [15] and analyzed the air gap flux density distribution and flux density of the rotor and stator. However, the 2D equivalent model does not consider the skew effect due to the shape of claw-pole. Reference [16] used the equivalent magnetic network method to establish the equivalent magnetic circuit network model of the double-stator and single-rotor disk permanent magnet motor, used the arc-line method to obtain various leakage reluctances of the motor, and analytically calculated the gap flux leakage coefficient and the average air gap flux density to greatly shorten the calculation time. In [17], the permanent magnetic flux leakage and the equivalent magnetic circuit network are solved separately. Although the model is simplified, the calculation time is reduced, and the calculation accuracy is improved. Reference [18] proposes a simple nonlinear magnetic analysis of axial-flux permanent magnet motors as an aided design tool for 3-D finite element analysis (3D-FEA). The time is significantly reduced, but the influence of magnetic flux leakage is ignored, and the calculation accuracy is not high. Reference [19] established the equivalent magnetic circuit network model of this kind of motor and obtained the analytical expression of the air gap leakage coefficient of this kind of motor. The three-dimensional finite element method and the test of the prototype motor were used to verify the established equivalent correctness of the magnetic circuit network model and the calculation of the air gap flux leakage coefficient. In [20], a novel analytical model combining the subdomain approach and the magnet equivalent circuit approach was proposed to predict the cogging rotation of a surface mount permanent magnet motor with rotor eccentricity and magnetic defects for any combination of slot and pole. Reference [21] established a three-dimensional equivalent magnetic network model based on the complex magnetic circuit of an axial-flux hybrid excitation motor. The air-gap flux density, flux linkage, back EMF, torque, and fault tolerance of the motor are solved by this model and compared with 3-D finite element analysis (FEA) to verify its validity and correctness. Reference [22] analyzes the conditions that must be met to eliminate the cogging torque of the induced wave through a direct, accurate, and original method, thereby reducing the cogging torque. This method does not require finite element method iterations to achieve the desired accuracy. Although the long-term calculation of the finite element method is avoided, its original basic analysis has a large error and a large workload.

The holding torque and detent torque of the claw-pole stepper motor are very important, which not only affect the performance of the motor, but also affect the accuracy of the position control [23].Holding torque and positioning torque of the claw-pole stepper motor are very important, which not only affect the performance of the motor, but also affect the accuracy of the position control [20]. In order to better study the performance optimization of the motor and reduce the cost of optimization time, based on the literature [14], this paper adopts the method of combining the equivalent magnetic circuit method and the three-dimensional finite element analysis to analyze the micro permanent magnet claw pole stepper. For the micro claw-pole stepping motor, which is the research object of this paper, the phenomena of magnetic flux leakage and magnetic saturation are particularly prominent, and the analysis is difficult. Many scholars have established a more accurate three-dimensional equivalent magnetic circuit after considering magnetic flux leakage and magnetic saturation [15,20,21]. However, this method presents difficulty in analysis and large errors. Therefore, this paper wants to avoid the difficult analysis in the equivalent magnetic circuit method through three-dimensional finite element simulation based on the equivalent magnetic circuit method. This can improve the accuracy of the equivalent magnetic circuit method, and also avoid the problem that it takes a lot of time to use the finite element method completely. In addition, most scholars use the finite element method to verify the accuracy of the models they build, or to verify the accuracy of the methods they use. On the contrary, it is ignored that the finite element method can also be combined with the equivalent magnetic circuit method, and the two methods are good at analyzing the motor design. Although this method also requires a certain amount of time to calculate, this method improves the accuracy of the model built, and given such a motor with a complex magnetic circuit, this computational time cost is acceptable. Moreover, before the finite element simulation, a simplified equivalent magnetic circuit model of the motor was established to address the complex magnetic circuit problem of the motor, and the air-gap flux equation was deduced, thereby deriving the relationship between the electromagnetic torque and the geometric parameters of the motor. Then, the 3D finite element simulation results are used to correct and verify the derived electromagnetic torque equation, before we use this method to carry out numerical optimization research on the air gap between the motor’s fixed rotation and the gap between the claw poles. On the premise that the detent torque does not change much, the holding torque is increased by 12.5%. The simulation calculation time is effectively shortened, the analysis difficulty is reduced, and the calculation accuracy is high. Finally, the experimental data, simulation data, and theoretical calculation values are compared and verified, which provides a reference for the design optimization of this type of motor.

## 2. Structure of a Micro Claw–Pole Stepper Motor

The rotor of the micro claw pole stepper motor studied in this paper is generally composed of a multi-pole permanent magnet material radially magnetized and a rotating shaft, to obtain a higher surface magnetic density. The permanent magnet material is NdFeB-n52 permanent magnet material. The stator consists of a concentric annular excitation coil and a yoke with claw teeth. As shown in Figure 1, the claw pole is divided into two sections along the axial direction, and the two sections are staggered by one quarter of the tooth pitch (90 electrical angle). Each segment of the two-phase motor consists of 10 claw teeth that mesh-up and down. The outer diameter of the micromotor is only 6 mm, and the step angle is 18°. Figure 2 shows a photo of the motor with a diameter of 6 mm (0.01 mm measurement error). Table 1 shows the specific parameter specifications of the model of the micromotor analyzed in this paper. As shown in Figure 3, in order to better and more accurately measure the static torque of the motor, according to these parameters, the model of the motor is accurately established in the finite element simulation software Infolytica MagNet.

## 3. Equivalent Magnetic Circuit Method

The magnetic circuit of the micro claw-pole stepper motor is a three-dimensional magnetic circuit, and the main magnetic flux includes not only radial magnetic flux, but also axial magnetic flux. Moreover, its leakage flux is more complex, including magnetic flux leakage between claw poles and claw poles, between permanent magnets, and between permanent magnets and claw poles. Since the leakage flux of this kind of micro motor accounts for a large proportion of the total magnetic flux, and the ferromagnetic part of the stator is relatively easy to saturate, the complexity of the electromagnetic calculation of the motor is increased.

From the magnetic circuit cloud diagram of the finite element simulation in Figure 4, we can confirm that the magnetic flux of the micro permanent magnet claw-pole stepper motor starts from the permanent magnet rotor, flows into the claw-pole through the air gap, then flows into the yoke along the tooth axis, and then flows into the yoke along the tooth axis. It flows into the housing along the yoke, flows out along the other claw pole, and finally returns to the rotor to form a loop. Figure 5 is a schematic diagram of the magnetic flux path. The equivalent magnetic circuit model established by the magnetic flux of the motor based on the equivalent magnetic circuit theory is presented in Figure 6. Table 2 substitutes a different expression for each value of the miniature permanent magnet claw-pole stepper motor, and then uses these reluctances, the flux of the permanent magnet flux, and the flux of the stator side-wound windings to calculate. Hence, we get the desired magnetic flux density in the stator-rotor air gap. The accuracy of the equivalent magnetic circuit model depends on the accuracy of each reluctance calculation [21]. Most scholars will calculate the individual magnetoresistances based on the measured parameters. However, for the micro claw-pole stepper motor studied in this paper, the volume size is too small and the measurement error is large, which directly leads to the low accuracy of the reluctance calculation. In addition, due to the small size of the motor, the local magnetic circuit is saturated, which further increases the calculation error. Although the accuracy of calculating the value of each magnetoresistance through finite element simulation is high, it takes a lot of time to solve. Therefore, this paper uses the finite element method to correct the reluctance calculation of the studied motor, and then brings these relatively accurate values into the analytical formula to improve the accuracy of the equivalent magnetic circuit model. On the one hand, the time cost is greatly reduced, and on the other hand, the accuracy of the built model is improved.

## 4. Calculation of Motor Torque

For the micro permanent magnet claw-pole stepper motor, requirements for the miniaturization of the motor are becoming increasingly strict. At the same time, as the volume continues to decrease, small changes in various parameters can significantly change the static torque performance of the motor. From the literature [5], it is hoped that the micro permanent magnet claw-pole stepper motor can improve the holding torque on the premise of keeping the positioning torque within a certain range. Therefore, this section derives the holding torque and detent torque of the studied motor on the basis of the previous section.

The mathematical model of the holding torque of the miniature permanent magnet micro claw pole stepper motor can be derived by the magnetic co-energy increment method:(1)$$dW_{e}=dW_{f}+dW_{m}$$
(2)$$\left\{\begin{array}{c} dW_{e}=eidt=id\psi \\ dW_{m}=Td\theta \\ dW_{f}=\frac{1}{2}id\psi \end{array}\right.$$
where $W_{e}$ represents input electrical energy, $W_{f}$ represents the energy in the coil, $W_{m}$ represents the output mechanical energy. *θ* represents the angle that the motor turns. T represents the motor torque.

After calculation, the electromagnetic torque can be obtained:(3)$$T\left(\theta \right)=\frac{id\psi }{d\theta }-\frac{1}{2}\frac{id\psi }{d\theta }=\frac{1}{2}i_{1}{}^{2}\frac{dL_{11}\left(\theta \right)}{d\theta }+i_{1}i_{2}\frac{dL_{12}\left(\theta \right)}{d\theta }+\frac{1}{2}i_{2}{}^{2}\frac{dL_{22}\left(\theta \right)}{d\theta }$$
where $i_{1}$ is the stator coil current, $i_{2}$ is the rotor-side permanent magnet equivalent current, $L_{11}$ and $L_{22}$ are the self-inductance, and $L_{12}$ is the mutual inductance, because the self-inductance is not independent of the position of the rotor, so the first and third terms in the formula are zero. Hence, after simplified derivation, we can get:(4)$$T\left(\theta \right)=iNP\frac{d\phi_{g}\left(\theta \right)}{d\theta }$$

After fully considering its magnetic flux leakage, the improved equivalent magnetic circuit diagram can be obtained. After simplification, the simplified equivalent magnetic circuit diagram of Figure 3 can be obtained. According to the calculation, the main magnetic flux of the air gap can be obtained as:(5)$$\phi_{g}=\frac{\frac{F_{c}}{2}\cdot \frac{\frac{R_{t\sigma }}{2}}{\frac{R_{t\sigma }}{2}+R_{1}}}{\frac{R_{1}\cdot \frac{R_{t\sigma }}{2}}{R_{1}+\frac{R_{t\sigma }}{2}}+R_{2}+R_{m}}+\frac{R_{m}}{\frac{R_{1}\cdot \frac{R_{t\sigma }}{2}}{R_{1}+\frac{R_{t\sigma }}{2}}+R_{2}+R_{m}}\phi_{r}$$
where
$$\left\{\begin{array}{c} R_{1}=\left(R_{y}//R_{ty}\right)/2+R_{t}\\ R_{2}=\frac{R_{m}}{2}+R_{g}\\ R_{y}=2R_{y1}+2R_{y3}+R_{y2}+R_{0} \end{array}\right.$$

Putting Equation (5) into Equation (4), and assuming that two phases are energized at the same time, the simplified calculation can be obtained:(6)$$T\left(\theta \right)=\sqrt{2}\cdot P\cdot \phi_{g}\cdot N\cdot i\cdot sinP\theta$$

For the miniature permanent magnet micro claw pole stepper motor, there is a dislocation angle of *δ* (the research object in this paper is 18°) between the AB phases, so the torque also has the same dislocation angle. So, we get:(7)$$T_{A}\left(\theta \right)=\sqrt{2}\cdot P\cdot \phi_{g}\cdot N_{A}\cdot i_{A}\cdot sinP\theta$$
(8)$$T_{B}\left(\theta \right)=\sqrt{2}\cdot P\cdot \phi_{g}\cdot N_{B}\cdot i_{B}\cdot sinP\left(\theta +\delta \right)$$

So, the whole electromagnetic torque is:(9)$$T\left(\theta \right)=T_{A}\left(\theta \right)+T_{B}\left(\theta \right)=\sqrt{2}\cdot P\cdot \phi_{g}\cdot N\cdot i\cdot \left[sinP\theta +sinP\left(\theta +\delta \right)\right]$$

When there is no current, the formula of detent torque is as follows:(10)$$T_{detent}\left(\theta \right)=\frac{1}{2}\phi_{g}{}^{2}\frac{dR}{d\theta }$$
where *R* represents the total reluctance of the magnetic flux, $\phi_{g}$ represents the magnetic flux, and θ represents the angle that the motor turns.

The finite element analysis model established based on the actual measured value is simulated, and the final holding torque and detent torque are shown in Figure 7. The maximum holding torque of the finite element simulation value in the Figure 7 is 0.341 mN·m. In addition, the maximum holding torque calculated by the equivalent magnetic circuit method used in this paper is 0.370 mN·m. The maximum holding torque calculated by the method is 0.370 mN·m. The error between the two is only 8.5%, less than 10%, but the calculation time of the equivalent magnetic circuit method is only a few minutes, which is far less than the 6 h of calculation time of the finite element method. In addition, the detent torque results obtained by the finite element method and the equivalent magnetic circuit method are relatively close, and the error is small, which shows that the established equivalent magnetic circuit model is correct and has high accuracy.

In order to further verify the accuracy of the established model, the actual measurement of the holding torque of the motor studied in this paper is carried out. Moreover, the results obtained with EMCM calculation value and 3D FEA simulation are verified and compared. As shown in Figure 8, the actual measured maximum holding torque of this motor is 0.32 mN·m. The difference from the finite element simulation value is 0.021 mN·m, and the difference from the EMCM calculation value is 0.05 mN·m. The measured value is smaller than the simulated and calculated values. Considering the existence of friction in the actual measurement and the influence of the accuracy of the spring balance, it can be considered that the method in this paper is accurate. The calculated values are in good agreement with the experimental results.

## 5. Optimization of Design Parameter

On the basis of the research in the previous chapters, the static torque optimization of the motor studied is carried out. On the premise that the positioning torque does not change much, the holding torque of the motor is increased. In the deduction of the electromagnetic torque formula in the previous sections, it can be found that the air-gap reluctance and the leakage reluctance in the air-gap have a direct impact on the electromagnetic torque of the motor. In addition, the size of the air gap between the stator and rotor is very easily affected by the assembly error, so it is very meaningful to study the influence of the air gap between the stator and the rotor and the gap between the claw poles on the holding torque and positioning torque of the motor.

As shown in Figure 9a, under the condition that other conditions are guaranteed, we study the holding torque and detent torque under different air gaps from 0.1 mm to 0.5 mm. From the figure, it is clear that the holding torque under 0.1 mm air gap is larger than that under 0.30 mm. However, through the comparison of the magnetic density nephogram shown in Figure 10, although the 0.1 mm air gap can bring a large torque, a large magnetic saturation phenomenon occurs in the claw teeth, leading to distortion of the torque waveform. This air gap will affect the stable operation of the motor and easily causes jitter. At the same time, such a small air gap is not easy to achieve in actual production. We think that the air gap g between 0.30 mm is the optimal value, and the air gap within this range is also suitable for the tolerance of assembly errors in actual production. As shown in Figure 9b, on the premise that the air gap between the stator and rotor is 0.30 mm, this paper further studies the influence of different values of the gap w between the stator claw poles on the motor holding torque. When w is greater than or equal to 0.20 mm, although the holding torque can be increased to a certain extent, the waveform is distorted. Relatively speaking, the optimal situation is when the gap w between the claw poles is 0.15 mm.

To sum up, when the value of the air gap g between the stator and rotor of the optimized motor is 0.30 mm, and the value of the gap w between the claw poles is 0.15 mm, the holding torque of the motor is increased from the original 0.32 mN·m to 0.36 mN·m. As shown in Figure 11, on the premise that the detent torque does not change much, the holding torque is increased by 12.5%.

## 6. Experimental Validation

Because the torque of the micro permanent magnet claw-pole stepper motor is very small, the conventional measuring machine is not suitable for the torque test of the miniature motor studied in this paper. Therefore, a high-precision spring balance is used to measure the motor torque, as shown in Figure 12. The final motor torque measurement is:(11)$$T=\left(F-F^{\prime }\right)\times R$$

*T*/θ is the static characteristic of the relationship between the reaction torque and the angle at which the rotor deviates from the equilibrium position. The current was retained constant at 0.2 A throughout the test. Moreover, because this type of motor heats up very quickly and has a huge impact on the performance of the motor, during the measurement process, the test needs to be completed within 5 s of power-on, and the next measurement can be performed after the motor temperature reaches a stable state. Connecting both ends of the spring scale will affect the speed of reading the measurement data. Therefore, the method in Figure 13 is used, and a fixed weight is connected to one end of the line. In the actual measurement, only one end of the spring scale needs to be read to improve the measurement accuracy. Because the torque of the micro permanent magnet micro claw-pole stepper motor is affected by too many factors, to obtain a more accurate measurement value, attention should always be paid to the tension of the wire in the actual measurement process, and the spring balance and the fixed wire head must be on a vertical line. The second most important thing is that the coil can only be wound around once. Too many turns will make the measured value far smaller than the actual value. In addition, it is necessary to determine the rotation direction of the motor before measurement and verify that the winding direction is consistent with the rotation direction of the motor. As shown in Table 3, the test average value of the holding torque is 0.3250 (mN·m). Furthermore, the simulation value is 0.3494 (mN·m). The tested holding torque value is similar to the simulation value, which verifies the correctness of the above simulation results.

## 7. Conclusions

Based on the field-circuit combination method, the static torque characteristics of the miniature permanent magnet claw-pole stepping motor are studied by the method of combining the equivalent magnetic circuit method and the three-dimensional finite element analysis. Hence, the studied motor is optimized. Finally, the correctness of the method and the effectiveness of the optimal design are proven by the comparison and analysis of experimental data, simulation data, and theoretical calculation values. The final result shows that the calculation error of electromagnetic torque can be within 8.5% using the method in this paper. Secondly, for the optimized motor, the positioning torque does not change much, and the holding torque is increased by 12.5%. The method of combining the equivalent magnetic circuit method and the finite element method proposed in this paper can effectively save the calculation time on the premise of ensuring high calculation accuracy and is suitable for the electromagnetic and optimal design of this kind of motor. The theoretical analysis also has certain reference value.

## Figures and Tables

**Figure 1 micromachines-13-01517-f001:**
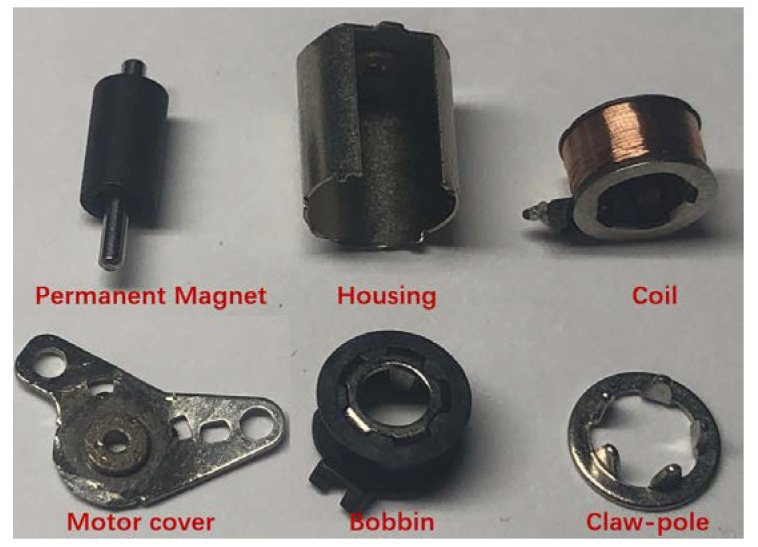
Structure diagram of a micro claw-pole stepper motor.

**Figure 2 micromachines-13-01517-f002:**
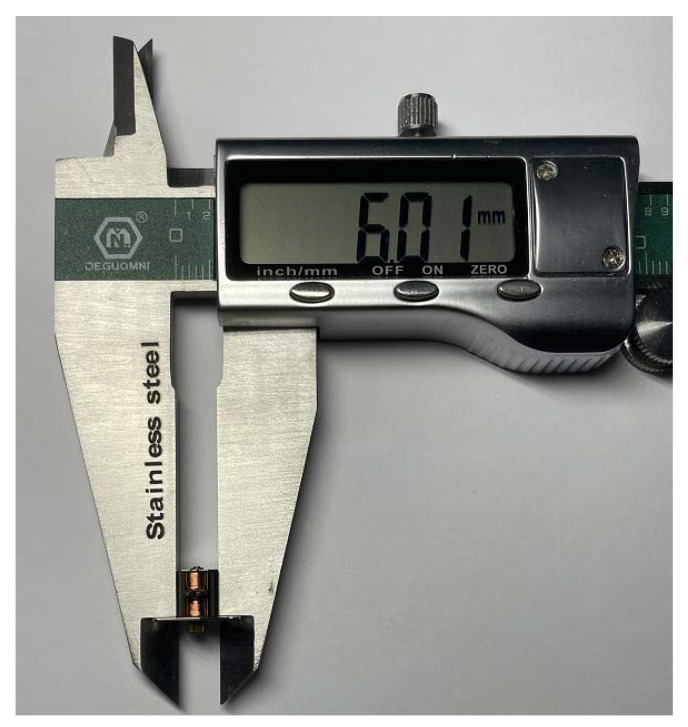
Photo of a 6 mm stepper motor.

**Figure 3 micromachines-13-01517-f003:**
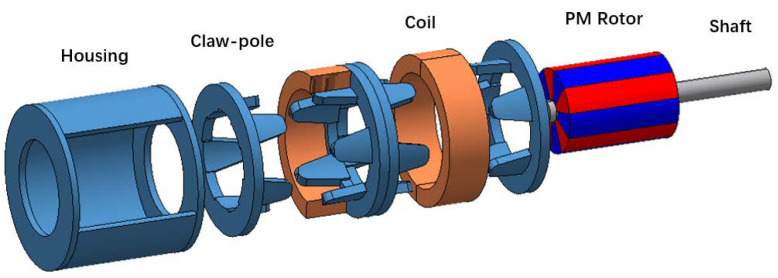
3D modeling diagram of a micro claw-pole stepper motor.

**Figure 4 micromachines-13-01517-f004:**
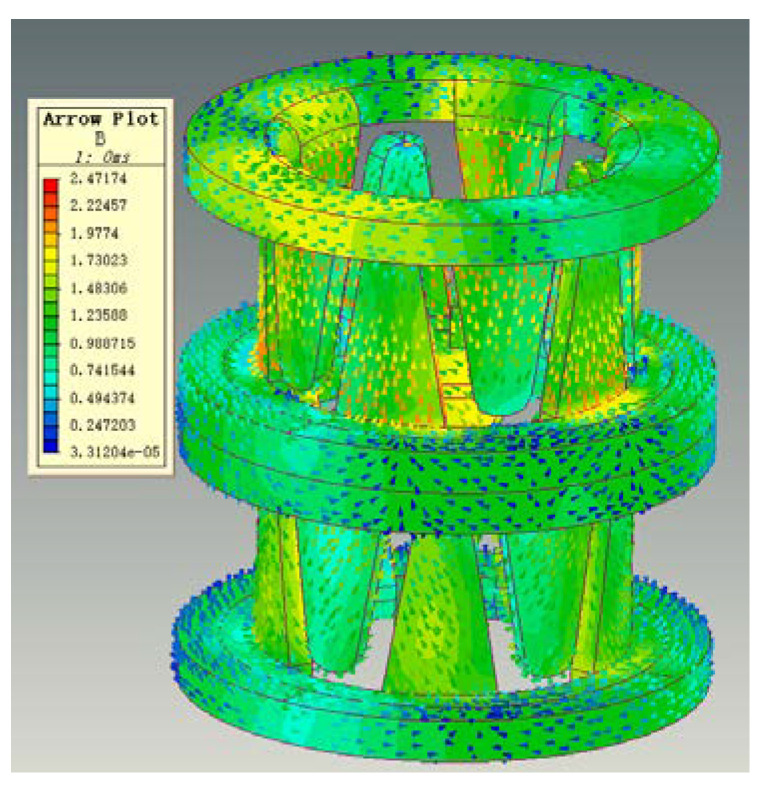
The magnetic circuit diagram of a micro claw-pole stepper motor.

**Figure 5 micromachines-13-01517-f005:**
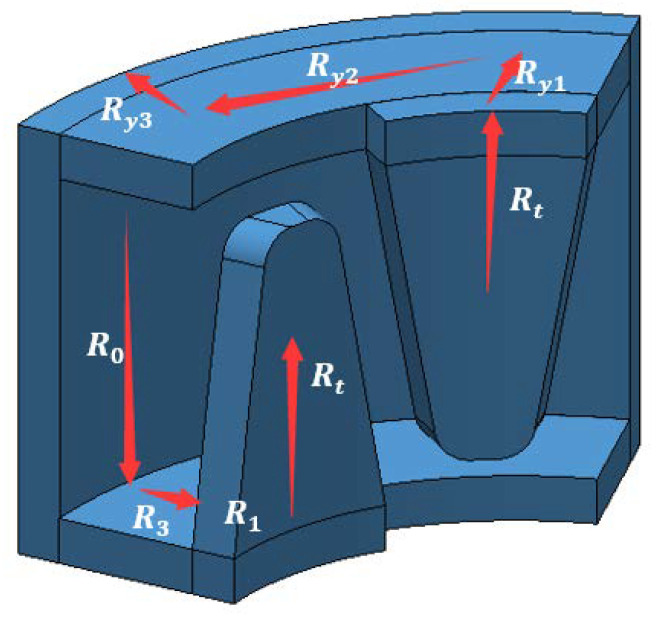
Schematic diagram of the magnetic flux path.

**Figure 6 micromachines-13-01517-f006:**
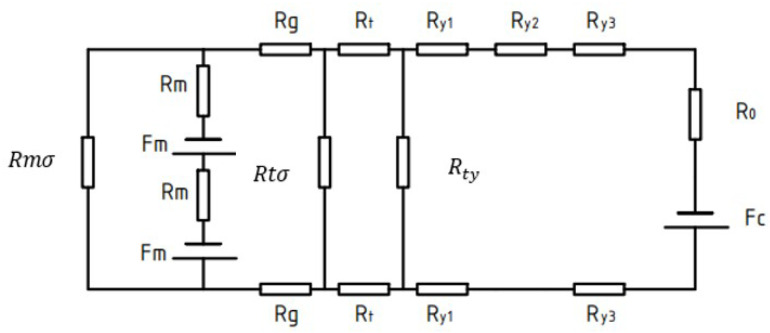
Equivalent magnetic circuit modeling.

**Figure 7 micromachines-13-01517-f007:**
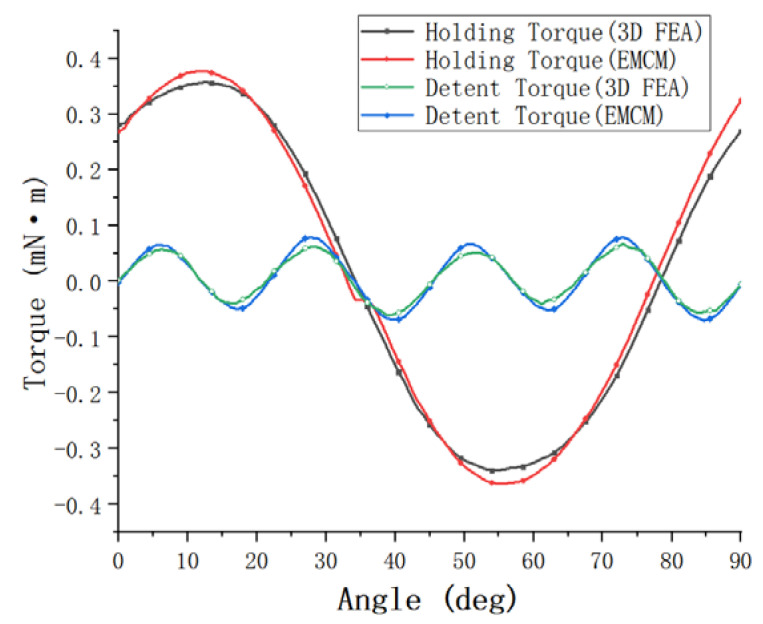
Equivalent magnetic circuit modeling.

**Figure 8 micromachines-13-01517-f008:**
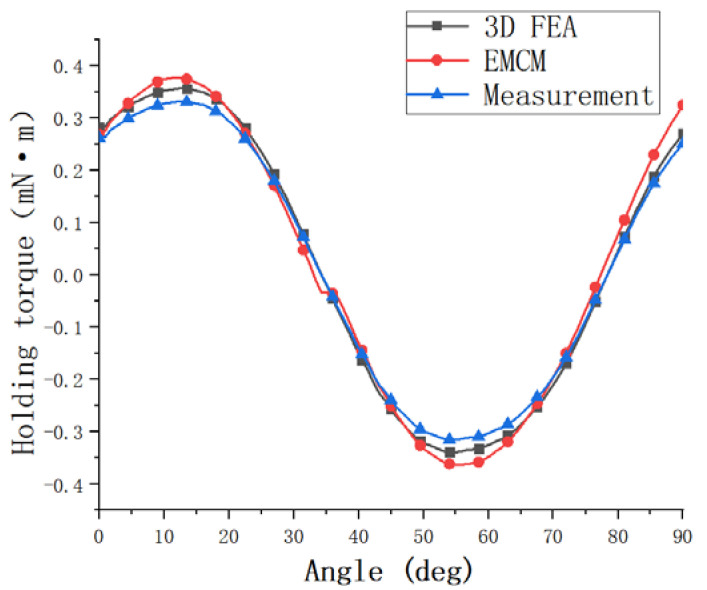
Holding Torque Comparison Chart.

**Figure 9 micromachines-13-01517-f009:**
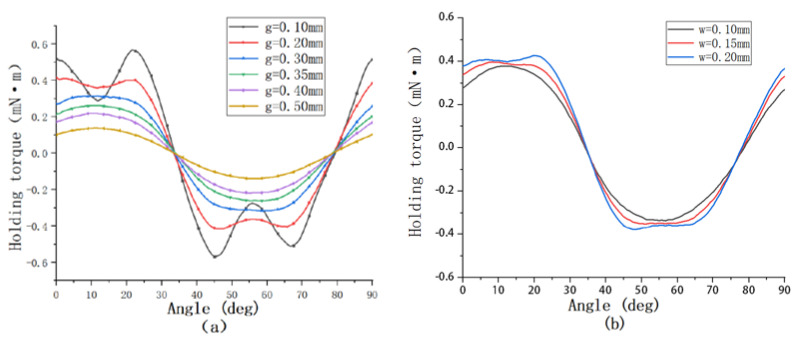
(**a**) Comparison of holding torque at different air gaps; (**b**) Comparison of holding torque at different gaps between claw poles.

**Figure 10 micromachines-13-01517-f010:**
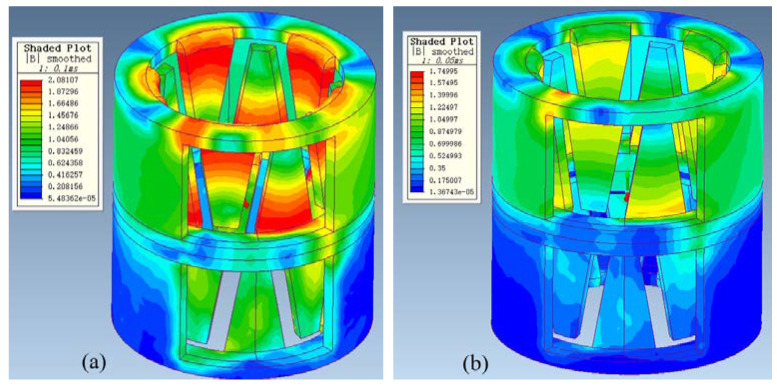
(**a**) Magnetic density cloud map under 0.10 mm air gap (**left**); (**b**) Magnetic density cloud map under 0.30 mm air gap (**right**).

**Figure 11 micromachines-13-01517-f011:**
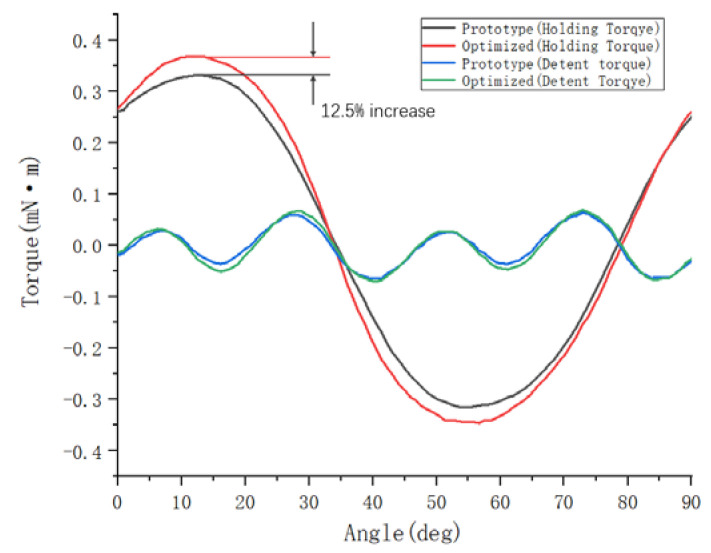
Comparison diagram before and after torque optimization.

**Figure 12 micromachines-13-01517-f012:**
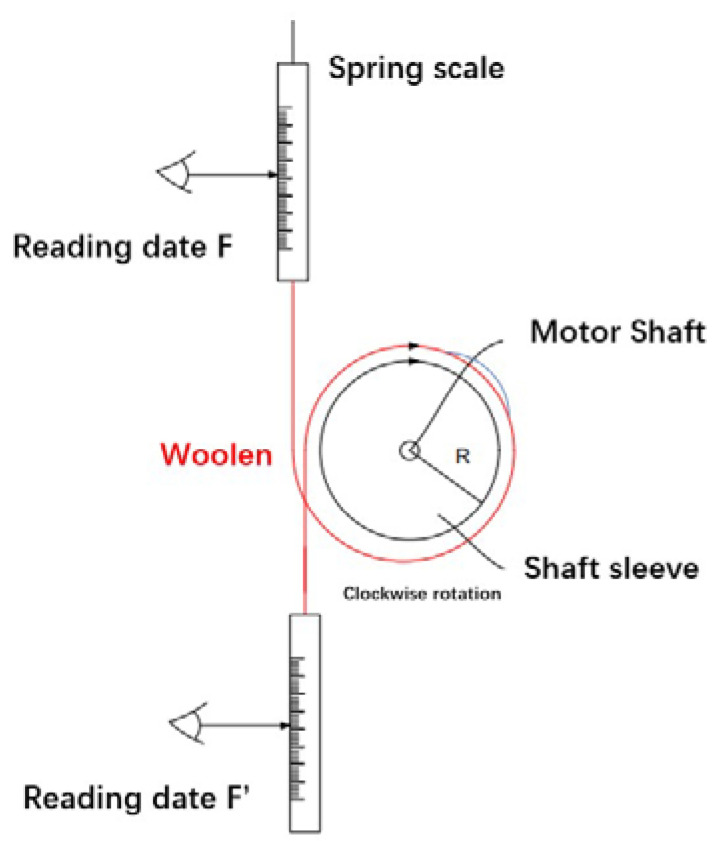
Structure diagram of torque test bench.

**Figure 13 micromachines-13-01517-f013:**
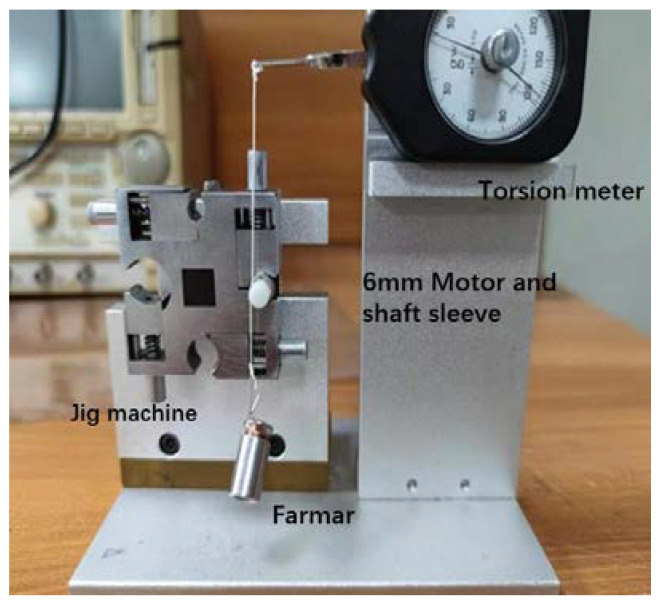
Micro stepper motor torque test bench.

**Table 1 micromachines-13-01517-t001:** Main parameter of prototype motor.

Parameter	Value	Parameter	Value
Number of phase	2	Claw-pole bottom thickness	0.15 mm
Number of claw poles	10	Claw-pole tip thickness	0.15 mm
Outer diameter of stator	6 mm	Claw-pole height	1.73 mm
Inner diameter of stator	3.2 mm	Air-gap length	0.2 mm
Axial length of stator	3.91 mm	Thickness of magnets	1.05 mm
Axial length of rotor	3.87 mm	Surface remanence	0.69 T
Thickness of housing	0.2 mm	Relative PM permeability	1.05
Thickness of claw-pole end-plate	0.2 mm	Number of turns per phase	143

**Table 2 micromachines-13-01517-t002:** Equivalent reluctance of each region.

Region	Explanation	Region	Explanation
$R_{t}$	Claw-pole resistance	$R_{m}$	PM resistance
$R_{y1}$	Claw-pole end-plate resistance	$R_{m\sigma }$	PM leakage resistance
$R_{y2}$	Yoke resistance	$R_{t\sigma }$	Claw-pole leakage resistance
$R_{y3}$	Yoke and Housing resistance	$R_{ty}$	Claw-pole and Yoke leakage resistance
$R_{0}$	Housing resistance	$F_{m}$	Permanent magnet magnetic potential
$R_{g}$	Air-gap resistance	$F_{c}$	Coil equivalent magnetic potential

**Table 3 micromachines-13-01517-t003:** Stepper motor holding torque test values.

Test Current *I*/*A*	Test Value (mN·m)	Holding Torque (mN·m)
0.2	0.342	0.352
0.2	0.339	0.341
0.2	0.327	0.362
0.2	0.302	0.353
0.2	0.315	0.339

## Data Availability

Not applicable.

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
