# Peer review of "Static Torque Analysis of Micro Claw-Pole Stepper Motor Based on Field-Circuit Combination"

_micromachines, 2022, doi:10.3390/mi13091517_

Round 1
Reviewer 1 Report (Previous Reviewer 1)
Accept the paper in its present form and the authors have a complete check for spelling mistakes and grammar.
Author Response
Thank you very much for your previous suggestions on this article, and it is a great honor to have your recognition of our work.
Reviewer 2 Report (Previous Reviewer 2)
The Authors have made remarkable modifications of this work and now it could be published in this journal.
Author Response
Thank you very much for your previous suggestions on this article, and it is a great honor to have your recognition of our work
Reviewer 3 Report (Previous Reviewer 3)
To analyze the static torque characteristics of a micro permanent magnet claw-pole stepper motor, the authors studied the method combining the equivalent magnetic circuit method and the three-dimensional finite element analysis method. Thank you for submitting the response letter for my comments. In the revised version, some of the reviewer’s requests were met by the authors. The reviewer would like to pay tribute to the authors’ work. However, I have some comments about the revised version. The reviewer’s comments are as follows:
---
>Comment #3:
The research survey is not enough. The articles listed in References are out of date. Besides, the authors should justify the effectiveness of the proposed method by comparing it with the latest methods. >Response:
It is true as the Reviewer suggested that our research survey is not enough. We conduct a large number of targeted research investigations. According to the revised comments, the content of this part has been rewritten.
>Comment to the authors’ response:
The research survey is still not enough. For example, the papers published within 3 years are used to calculate CiteScore in SCOPUS. Most of them are out of date. Therefore, this seems a well-behind article. The authors should survey past studies in detail. Besides, the authors should justify the effectiveness of the proposed method by comparing with the latest methods.
>Comment #5:
In the Introduction part, strong points of this proposed method should be further stated and the organization of this whole paper is supposed to be provided in the end.
>Response:
We have re-written this part according to the Reviewer’s suggestion. So we removed this section from the manuscript. We then conducted several targeted research investigations and rewrote this section as suggested. Reasonable changes have been made to the structure of the full text.
>Comment to the authors’ response:
I’m confusing. The reviewer did not request to remove the organization of the paper and the explanation about the advantage of the proposed technique. The authors should answer the reviewer’s comment appropriately.
>Comment #6:
The effectiveness of this work is not clear. Through simulations/experiments, the authors must justify the effectiveness of the proposed method by comparing it with the conventional methods. Several articles are discussed in the research survey. However, no comparison is shown with these techniques. Frankly speaking, the research survey and References are meaningless. The authors should show comparison data.
>Response:
We removed meaningless references and added relevant research studies in a targeted manner. As suggested, the advantages and disadvantages of several methods are compared, and experiments, data analysis, and comparative data are added.
>Comment to the authors’ response:
The authors should compare the proposed method with other researchers’ methods. In the revised version, the comparison was performed between EMCM and 3D FEA only. The authors should emphasize the difference with other researchers’ methods. Several articles are discussed in the research survey. However, no comparison is shown with these techniques. Frankly speaking, the research survey and References are meaningless.
>Comment #7:
The authors' proposed method does not adequately describe their data. The limitation and drawbacks of the existing techniques described in the introduction part are not supported by any demonstrations in section 6. Readers will fail to understand the scientific contribution of this research.
>Comment to the authors’ response:
We have corrected it according to the Reviewer’s comments. As suggested, the advantages and disadvantages of the prior art are described in detail, and experiments, data analysis, and comparative data are provided. The entire article has been reworked accordingly.
>Comment to the authors’ response:
I’m confusing the authors’ response. Although the authors described that “As suggested, the advantages and disadvantages of the prior art are described in detail, and experiments, data analysis, and comparative data are provided”, no quantitative data was shown in section 6. For example, the authors described that “References [16-19] studied and optimized the equivalent magnetic circuit model of the axial flux motor. The equivalent magnetic circuit method improves the accuracy of the analytical model, but the process of establishing and solving the equivalent magnetic circuit network also becomes complicated.” However, the authors' proposed method does not adequately describe their data. This interpretation is not supported by any demonstrations.
Author Response
Dear reviews
Exactly according to your comments, and found these comments are very helpful. I hope this revision can make my paper more acceptable. The revisions were addressed point by point below.
#1 Response to comment:(1) The research survey is still not enough. For example, the papers published within 3 years are used to calculate CiteScore in SCOPUS. Most of them are out of date. Therefore, this seems a well-behind article. The authors should survey past studies in detail. Besides, the authors should justify the effectiveness of the proposed method by comparing with the latest methods. (2) In the Introduction part, strong points of this proposed method should be further stated and the organization of this whole paper is supposed to be provided in the end.
Response: We have corrected it according to the Reviewer’s comments. In the Introduction section, we update several references within the last three years and demonstrate the effectiveness of the proposed method with a comparative description with state-of-the-art methods, as suggested by the reviewers.
#2 Response to comment: The authors should compare the proposed method with other researchers’ methods. In the revised version, the comparison was performed between EMCM and 3D FEA only. The authors should emphasize the difference with other researchers’ methods. Several articles are discussed in the research survey. However, no comparison is shown with these techniques. Frankly speaking, the research survey and References are meaningless.
Response: Based on the reviewers' suggestions, we have made appropriate modifications. In this paper, we hope to express the way we want to combine fields and paths through the comparison between EMCM and 3D FEA, select the advantages of EMCM and 3D FEA, and avoid the shortcomings of the two. Finally, higher calculation accuracy can be obtained, and the calculation time can be greatly shortened. In the introduction section, we have revised the description of this section.
#3 Response to comment: Although the authors described that “As suggested, the advantages and disadvantages of the prior art are described in detail, and experiments, data analysis, and comparative data are provided”, no quantitative data was shown in section 6. For example, the authors described that “References [16-19] studied and optimized the equivalent magnetic circuit model of the axial flux motor. The equivalent magnetic circuit method improves the accuracy of the analytical model, but the process of establishing and solving the equivalent magnetic circuit network also becomes complicated.” However, the authors' proposed method does not adequately describe their data. This interpretation is not supported by any demonstrations.
Response: Firstly, we added quantitative data analysis in Chapter 6 to demonstrate the validity of the experimental approach based on the reviewers' comments.Secondly, we revise the problems of references [16-19], describing and analyzing the methods they use.
Round 2
Reviewer 3 Report (Previous Reviewer 3)
In this paper, the authors studied the static torque characteristics of the micro permanent magnet claw pole stepper motor by using equivalent magnetic circuit 12 method (EMCM) and 3D finite element analysis (3D FEA). The key idea of the proposed technique is the combination of EMCM and 3D FEA. In the first version, due to poor research survey and lack of comparison data, the effectiveness of the proposed technique was not clear. However, in the revised version, most of the reviewer’s requests were met by the authors. The reviewer would like to pay tribute to the authors’ great work. This is scientifically sound and contains sufficient interest to merit publication, I think.
This manuscript is a resubmission of an earlier submission. The following is a list of the peer review reports and author responses from that submission.
Round 1
Reviewer 1 Report
Most of the references referred to are published before 2010, this section must be improved, identify the recently published relevant works, and improve the introduction section and literature review part of the paper.
Improve the quality of figures 5, 6, and 7, it is very difficult to read and understand
in the table.2 some parameters are confusing like whether they are indicated in the suffix or not
the authors need to describe their contribution clearly, so that novelty of the work may gives interest to the readers
Reviewer 2 Report
The paper concerns some specific engineering problem of the micro claw-pole stepper motor with coupled electromagnetic and mechanical effects.
The main objection behind the publication of this paper in any journal would be a complete lack of the Finite Element Method model and analysis description.
The Authors try to optimize this problem, but there is no precise description of the optimization method applied - this must be changed.
Finally, the experimental part is free from both experimental error analysis and discussion as well as from statistical description of the resulting parameters.
Reviewer 3 Report
In this paper, the authors studied the static torque characteristics of the micro permanent magnet claw pole stepper motor by using equivalent magnetic circuit 12 method (EMCM) and 3D finite element analysis (3D FEA). The key idea of the proposed technique is the combination of EMCM and 3D FEA. Overall, the authors have made a good attempt. However, due to poor research survey and lack of comparison data, the effectiveness of the proposed technique is not clear. The reviewer’s other comments are as follows:
1. The proposed presentation does not respect the elementary rules of a scientific writing. Some of the articles listed in References are not quoted in sentences. These articles are meaningless. The authors must check the manuscript before submitting this paper.
2. The authors must quote the references according to the reference number. (You must start the quotation according to the reference number.)
3. The research survey is not enough. The articles listed in References are out of date. For example, the papers published within 3 years are used to calculate CiteScore in SCOPUS. (Of course, only Ref. [29] is a new article. However, this is not essential, because the citation of this paper is that “… or use the experimental design 69 analysis to numerically optimize some variables of the motor [25-31].” See p. 2. There is no need to quote Ref. [29].) The authors should survey past studies in detail. Besides, the authors should justify the effectiveness of the proposed method by comparing with the latest methods.
4. The problem definition of this work is not clear. In the introduction part, the drawbacks of each conventional technique should be described one by one. The explanation of the existing techniques is rough. For example, “Many scholars also use the 2D equivalent model to analyze the motor or use the EMCM method to analyze the torque, back electro-motive force, and noise and vibration of the motor [21-24], or use the experimental design analysis to numerically optimize some variables of the motor [25-31]”. The authors should emphasize the difference with other methods to clarify the position of this work further.
5. In the Introduction part, strong points of this proposed method should be further stated and organization of this whole paper is supposed to be provided in the end.
6. The effectiveness of this work is not clear. Through simulations/experiments, the authors must justify the effectiveness of the proposed method by comparing with the conventional methods. Several articles are discussed in the research survey. However, no comparison is shown with these techniques. Frankly speaking, the research survey and References are meaningless. The authors should show comparison data.
7. The authors' proposed method does not adequately describe their data. The limitation and drawback of the existing techniques described in the introduction part is not supported by any demonstrations in section 6. Readers will fail to understand the scientific contribution of this research.